# Federated Meta-Learning with Attention for Diversity-Aware Human Activity Recognition

**DOI:** 10.3390/s23031083

**Published:** 2023-01-17

**Authors:** Qiang Shen, Haotian Feng, Rui Song, Donglei Song, Hao Xu

**Affiliations:** 1College of Computer Science and Technology, Jilin University, Changchun 130012, China; 2School of Artificial Intelligence, Jilin University, Changchun 130012, China; 3Chongqing Research Institute, Jilin University, Chongqing 401123, China

**Keywords:** human activity recognition, federated learning, meta learning

## Abstract

The ubiquity of smartphones equipped with multiple sensors has provided the possibility of automatically recognizing of human activity, which can benefit intelligent applications such as smart homes, health monitoring, and aging care. However, there are two major barriers to deploying an activity recognition model in real-world scenarios. Firstly, deep learning models for activity recognition use a large amount of sensor data, which are privacy-sensitive and hence cannot be shared or uploaded to a centralized server. Secondly, divergence in the distribution of sensory data exists among multiple individuals due to their diverse behavioral patterns and lifestyles, which contributes to difficulty in recognizing activity for large-scale users or ’cold-starts’ for new users. To address these problems, we propose DivAR, a diversity-aware activity recognition framework based on a federated Meta-Learning architecture, which can extract general sensory features shared among individuals by a centralized embedding network and individual-specific features by attention module in each decentralized network. Specifically, we first classify individuals into multiple clusters according to their behavioral patterns and social factors. We then apply meta-learning in the architecture of federated learning, where a centralized meta-model learns common feature representations that can be transferred across all clusters of individuals, and multiple decentralized cluster-specific models are utilized to learn cluster-specific features. For each cluster-specific model, a CNN-based attention module learns cluster-specific features from the global model. In this way, by training with sensory data locally, privacy-sensitive information existing in sensory data can be preserved. To evaluate the model, we conduct two data collection experiments by collecting sensor readings from naturally used smartphones annotated with activity information in the real-life environment and constructing two multi-individual heterogeneous datasets. In addition, social characteristics including personality, mental health state, and behavior patterns are surveyed using questionnaires. Finally, extensive empirical results demonstrate that the proposed diversity-aware activity recognition model has a relatively better generalization ability and achieves competitive performance on multi-individual activity recognition tasks.

## 1. Introduction

The task of human activity recognition targets inferring the semantic meaning of the situation based on the features extracted from sensor or vision signals. In recent years, a rich body of intelligent applications, including aging care [1,2], smart homes [3], fitness tracking [4], and sleep monitoring [5], have benefited from the automatic and unobtrusive recognition of users’ activity. A commonly used approach is sensor-based activity recognition, which utilizes on-body or ambient sensors to identify an individual’s contextual information details using machine learning models [6]. Existing implementations are quite diverse, from shallow models such as k-nearest-neighbors (kNN) [7] to deep neural networks such as multiple-layer perceptron (MLP) [8,9] and convolutional neural networks (CNN) [10].

Activity recognition involves collecting and processing personal behavior data for training purposes, which has important consequences in terms of data privacy. This has been addressed with Federated Learning (FL), an emerging machine learning technology that enables distributed learning of a global prediction model without compromising privacy [11]. Given a group of users, FL approaches [12] make use of local, user-specific supervision to update a global, high-quality activity predictors meant to be applicable to all users.

Despite the success of deep learning and federated learning for accurate activity recognition, there is one major limitation to the deployment of context-aware systems in the open world, which is the distribution discrepancy in sensor data across multiple individuals [13]. In real-world scenarios, sensor data are collected from a group of diverse individuals, and behavior patterns are person-dependent [14] owing to biological and environmental factors, meaning that the same activity can be performed differently by different individuals [15]. This presents a challenge for activity recognition tasks. For instance, individuals walk, eat, or interact with their phones in different manners, owing to psychological and biological factors. In addition, the diversity of lifestyles is also a major barrier to accurate activity recognition, which constantly happens in the real world. For instance, some people go to work by car, and some may prefer to walk to the office. Specifically, for ubiquitous computing systems, it is challenging to apply activity recognition models learned on existing users for predicting the activity of new-coming users with different characteristics and behavioral patterns. For machine learning algorithms, the heterogeneity of the sensory data input into the machine learning model contributes to the fact that training data and test data are not independent and identically distributed (i.i.d.) [13]. Thus, the performance of the machine learning model drops due to the heterogeneity of sensor data.

To tackle the heterogeneity challenge in activity recognition, domain adaptation techniques such as transfer learning and multi-task learning have been applied to transfer knowledge across different individuals. The authors of [16] proposed an HAR model with a particular layer with few parameters inserted between every two user-dependent layers of the CNN for personalization. The authors of [17] proposed to personalize their models with transfer learning. The authors of [18] proposed a personalized HAR model based on multi-task learning techniques, where each task corresponds to a specific person. However, this work is limited because it is impractical to train a personalized model for each user in real life, which is both extremely time- and energy-consuming. Some FL-based approaches handle cross-individual diversity by learning user-specific models [12,19,20]. Most importantly for our contribution, Meta-HAR [21] trains a shared embedding network in a federated manner and then adapts the network with an output layer to specific users via fine-tuning. However, these approaches ignore the problem of feature-level discrepancies.

Moreover, existing approaches tend to personalize a model for each specific individual. However, there are two main problems with activity recognition in the real world: (i) the performance of the model drops when the dataset of one single individual is limited and (ii) it is impractical to train the model for each user for a context-aware application with a large number of users to recognize their activity, especially for a new user. In addition, individual characteristics (e.g. personality) contributing to the heterogeneity in sensory data have never been explored or utilized to improve the generalization ability of the machine learning model. Previous studies have shown that human behavior has correlations with personality [22,23,24] and psychological factors [25,26]. Thus, we take these characteristics into consideration for clustering individuals to tackle this issue for the purpose of improving the generalization of the activity recognition model.

In this paper, we address the issue of heterogeneity by proposing DivAR, a federated meta-learning framework with an attention mechanism to learn a diversity-aware model adapting to the individuals’ discrepancies. Specifically, (i) we cluster the individuals who have the same characteristics from multiple aspects, including social factors (e.g. basic information, personality, and mental state) and behavioral characteristics. (ii) We apply a federated meta-learning framework to train a diversity-aware model accounting for both diversity and similarity for each cluster. Specifically, we train a meta-learning model in a federated learning manner, where a centralized meta-model learns common feature representation that can be transferred across all clusters of individuals, and multiple cluster-specific models stored in decentralized clients are utilized to learn cluster-specific features. (iii) In order to learn feature-level inter-individual discrepancies, we apply a CNN-based attention module to extract cluster-specific features from the global model for each client model. To evaluate the approach, we construct two diversity-aware activity recognition datasets, including sensor readings from smartphones with activity annotations using the data collected from multiple individuals in the real-life environment. In addition, the social characteristics such as personality and psychological factors of multiple individuals are surveyed by a set of questionnaires. We then explore the impact of the heterogeneity phenomenon in sensory data by setting an empirical experiment and analyzing the distribution of features. Finally, we conduct experiments on our dataset to evaluate the performance of the model. The results demonstrate that our diversity-aware model is able to achieve more accurate multi-individual activity recognition than state-of-the-art approaches, which indicates the promise of our proposed method. The main contributions of this work are summarized as follows:We propose a federated meta-learning framework for activity recognition, a centralized embedding network to extract shared sensory features, and specific features for a certain group of individuals are learned by the attention module in each decentralized model. In this way, both shared features and distribution discrepancy of sensory features are considered to handle heterogeneity challenges and ’cold-start’ problems for context-aware applications.The proposed model allows the model to be trained locally, and only the updates of the parameters are transferred across clients, which provides the ability to preserve privacy for activity recognition tasks in multi-individual or cross-organization scenarios.We explore multiple diversity factors across individuals for the activity recognition task. Various properties of an individual for clustering are considered, including characteristics such as personality and mental state in the activity recognition task to tackle the heterogeneity problem to achieve diversity-aware activity recognition.We construct two multi-individual heterogeneous datasets by collecting sensor readings from naturally used smartphones annotated with activity information in a real-life environment. In addition, social characteristics, including personality, mental health state, and behavior patterns, are collected for clustering and analyzing individuals from diverse aspects.We conduct experiments on multi-individual heterogeneous datasets. The experimental results indicate that the proposed federated meta-learning activity recognition model has a relatively competitive performance in terms of both handling heterogeneity on existing users and adapting to new users.

The remainder of the paper is structured as follows. The next section introduces the motivation for our proposed model. Section 2 positions DivAR with respect to existing approaches. Section 4 introduces DivAR, the proposed federated meta-learning framework for diversity-aware activity recognition. Then, Section 5 describes the procedure of data collection, and Section 6 describes and discusses our experimental evaluation of DivAR on real-world data. Finally, Section 7 presents some concluding remarks and illustrates promising directions for future work.

## 2. Related Work

### 2.1. Activity and Activity Recognition

As a promising research topic, human activity recognition (HAR) has produced extensive research in the areas of ubiquitous computing and machine learning [6]. Generally, the procedure of human activity recognition can be regarded as a standard time series classification problem by utilizing machine learning methods. A rich body of research on HAR has been carried out in simulated and controlled environments. In earlier years, shallow machine learning models were applied to activity recognition tasks to extract features manually. These models depended on statistical features [27] and distribution-based features [28] and then inferred the activity using different classification models. The authors of [29] proposed algorithms to recognize physical activities using the data collected using wearable sensors worn simultaneously on the body. The work in [30] investigates effective extractions of universal features from various sensors for HAR methods supporting context-aware applications.

Recently, deep learning [31] has been extensively applied in HAR [6]. For instance, CNN has shown great potential for extracting various features from sensory readings. The work in [10] proposed a CNN-based HAR model to extract the temporal features and spatial dependency over multiple sensors. The authors of [32] applied deep CNN to learn the representation of electrocardiogram (ECG) signals for the classification layers. Furthermore, DeepConvLSTM [33], a model integrating CNN and RNN, has shown notable performance in capturing spatial-temporal features from the sensor signal data. DeepSense [34] integrates CNN and RNN, which can not only model temporal dependencies but fuse the features capturing multimodal sensor readings.

However, these HAR algorithms classify sensor data into one certain labeled activity, which provides limited information about humans for context-aware applications. Moreover, these HAR methods are not suitable for real-world problems since most of them are implemented in controlled environments. Therefore, the study and recognition of activity in the open world have drawn extensive attention in the area of ubiquitous computing. Ref. [35] collect labeled data of natural behavior using smartphone and smartwatch sensors in real-life settings and then recognize the activity of individuals in the wild. The work in [36] treats the task of activity recognition as a multi-label classification problem and uses MLP as the classifier, which improves accurate recognition with fewer parameters. Ref. [37] proposes a multimodel CNN model for fusing diverse modalities of sensor readings to learn behavioral activity. DeepContext [38], a CNN-based model for recognizing a user’s activity, applies an attention mechanism to discover and utilize important features of smartphone sensor readings. However, previous work has hardly considered multiple aspects of activity and studied the correlations between aspects of the activity.

### 2.2. Handling Heterogeneity of Sensory Data

Many state-of-the-art approaches for human activity recognition assume that the training data and the test data are independent and identically distributed (i.i.d.). However, this is impractical since sensory data for activity recognition is heterogeneous. Behavior patterns are person-dependent [14] owing to biological and environmental factors, which means that the same activity can be performed differently by different individuals.

In practical human activity recognition scenarios, while a certain number of participants’ data can be collected and annotated for training, the target users are usually unseen by the systems [13]. Thus, the distribution divergence between the training data and the test data appear as a challenge in human activity recognition, especially for the recognition of complex and multi-aspect activities in the wild. There are some studies focusing on solving the heterogeneity challenge. The authors of [14] compare the universal and personal models, and the results indicate that the personal models perform dramatically better than the universal model. The authors of [39] studies the HAR task in diverse user populations and proposes a novel scalable activity classification framework to handle the increasing number of users.

Recently, personalized deep learning models for heterogeneity between users are widely applied in activity recognition tasks. The authors of [40] proposed a personalized HAR approach for each individual in a multi-person environment. The work in [16] learned user-specific parameters of a CNN for each user using a small amount of data. The authors of [17] proposed to personalize CNN models with transfer learning by training the models with data collected from a few participants and then only fine-tuning the top layers of the CNN with a small amount of data for the target users. The authors of [41] defined and utilized the discrepancy and consistency across individuals on the task of HAR for mobile sensing applications. However, existing work has hitherto ignored the analysis of heterogeneous data and the factors contributing to the heterogeneity in sensor data.

Multi-task learning [42] has been used successfully across all applications of machine learning, from natural language processing [43] and speech recognition [44] to computer vision [45]. As a classic type of transfer learning, multi-task learning aims to leverage information contained in several related tasks in order to improve the performance of all the tasks [42]. Essentially, MTL aims at improving the generalization ability of the model, when the multiple tasks are sufficiently related [42]. In addition, MTL is applied to solve the scenario where data are unlabeled. Tang et al. [46] propose a semi-supervised model that effectively learns to leverage unlabeled mobile sensing datasets to complement small labeled datasets with multi-task self-supervision.

As mentioned above, machine learning tasks such as activity and activity recognition are facing a heterogeneity problem. Namely, the one-size-fits-all machine learning models cannot perform equally well for each user given the cross-individual differences. In recent years, therefore, a few studies have proven that domain adaptation techniques show potential in handling heterogeneity problems for sensor-based recognition tasks such as HAR [47] and wellbeing recognition [48]. By improving the generalization of the estimated models for multiple related learning tasks via capturing the tasks’ relationships, MTL has been used for solving the heterogeneity challenges in the ubiquitous computing field. It has been theoretically and empirically shown to be more effective than learning tasks individually. In addition, considering privacy issues and annotation costs, Saeed et al. [49] propose a self-supervised technique for feature learning from sensory data.

The work in [50] proposed a personalized machine learning for robot perception of affect and engagement in autism therapy, which outperforms one-size-fits-all ML approaches. The authors of [48] applied a personalized multitask learning technique on three wellbeing prediction models and proved that the personalization of the model can improve the performance of both shallow and deep models. In addition, multi-task learning has also been used to learn models for related tasks in ubiquitous computing areas. Aroma is proposed in [51] to train a deep multi-task learning model for the joint recognition of simple and complex human activity.

However, these studies require a large amount of data for a deep learning model, and the factors contributing to the heterogeneity have never been explored and utilized. While our work can consider both behavioral diversity and similarity in the activity recognition model and guarantee that knowledge can be transferred between individuals.

### 2.3. Federated Learning

Federated learning aims to train a centralized model using the data stored in multiple distributed nodes in a privacy-aware manner [52]. Federated Averaging [53] combines local stochastic gradient descent (SGD) on client nodes with model averaging on the server side and is able to reduce communication rounds between clients and servers. Existing studies [53,54] have shown that federated learning performs well when clients hold non-IID data and thus has some potential for addressing cross-individual diversity in HAR [55]. However, although federated learning-based HAR approaches succeed in learning from different clients, pure FL does not model the similarity and discrepancies of the clients and thus fails to learn personalized models for all individuals. MTL [42] combines information from multiple, related learning tasks to improve the prediction performance of all the tasks simultaneously and represents a natural strategy for dealing with cross-individual differences, and exploiting cross-individual similarities, in HAR [56]. MOCHA is proposed as a general federated multi-task learning framework and performs well for HAR tasks. However, it ignores the case of heterogeneous data distributions. Most closely related to our approach, Meta-HAR [21] solves personalized HAR by treating each individual as a separate task and learning both shared and user-specific information. Compared with learning a one-size-fits-all model, MTL approaches can precisely capture relationships among non-IID data and are naturally well-suited for dealing with user heterogeneity in cross-individual HAR. However, existing works design the MTL model by taking a feed-forward network and splitting the network at the *classification* layer, therefore ignoring the discrepancy at the feature level.

### 2.4. Meta-Learning

Meta-learning aims to solve the problem of learning to learn on a wide range of learning tasks [57]. Andrychowicz et al. [58] adopt deep neural networks to train a metalearner and propose an optimizer–optimizee setup, where each component is learned with an iterative gradient-descent procedure. Model-agnostic meta-learning (MAML) [59] is another popular approach that does not impose a constraint on the architecture of the learner. Ravi and Larochelle [60] propose an LSTM meta-learner to learn an optimization procedure for few-shot image classification. Li et al. [61] develop an SGD-like meta-learning process and also experiment on few-shot regression and reinforcement learning problems. Reptile [62], i.e., the approach adopted in this paper, simplifies the learning process of MAML by conducting first-order gradient updates on the meta-learner. Jiang et al. [62] interpreted federated learning as an MAML algorithm and implement a federated version of the first-order MAML algorithm, Reptile. However, Jiang et al. [62] focus on parameter tuning to obtain a global model which is readily personalized. Wijekoon et al. [63] proposed to learn personalized models for individuals to improve adaptation ability by meta-learning. Existing studies combine federated learning with meta-learning. Chen et al [64] propose FedMeta, a Federated Meta-Learning to handle the statistical distribution and systematic challenge in the real-world application of federated learning. Fallah et al. [65] propose learning personalized models using meta-learning and federated learning to train models across multiple computing units. Li et al. [21] introduce Meta-HAR, a federated representation learning framework treating activity recognition tasks for each individual as a related task and applying meta-learning to learn personalized models.

## 3. Motivation

The heterogeneity of data is the major barrier to a high-quality machine learning model, especially heterogeneity with users will decrease the performance of the model when applied in the open world [35]. In this section, an empirical experiment is conducted to prove that heterogeneous sensory data decrease the performance of the machine learning model for activity recognition using the in-the-wild sensory data to motivate the diversity-aware activity recognition model.

### 3.1. Feature Visualization

We firstly explore the distribution of features extracted from in-the-wild sensory data. The information on datasets will be introduced in Section 5. In order to visualize the features, a Kernel Density Estimator (KDE) is used to show both the marginal and conditional distribution of each feature extracted from multi-modal sensor readings. The KDE function can be formally defined as:$${\hat{p}}_{n}\left(x\right)=\frac{1}{nh}\sum_{i=1}^{n}K\left(\frac{x_{i}-x}{h}\right)$$
where K(x) is the kernel function that is generally a smooth function, the Gaussian kernel is used in this experiment, and h>0 is the smoothing bandwidth that controls the amount of smoothing. Basically, the KDE smoothes each data point $X_{i}$ into small density bumps and then sums all these small bumps together to obtain the final density estimate.

Figure 1 shows the distribution of features for all instances of multiple users. Specifically, each subplot in Figure 1 indicates the estimation curves of a single feature, where each curve refers to one single user. For better visualization, 8 individuals out of 30 are randomly selected for plotting the distribution. In most of the subplots, features extracted from different individuals’ sensory data represent remarkably different distributions. The discrepancy across individuals is partially caused by the different behavioral styles of performing one certain activity. For some certain features, such as the variance of rotation vectors and accelerometers, however, the distributions are quite similar.

### 3.2. Impact of Heterogeneous Sensory Data on Activity Recognition

In this subsection, we analyze the inter-individual differences and their impact on the activity recognition model. Behavioral diversity indicates that each individual has their own way of behaving and thus the same activity can be performed differently by different individuals. Essentially, the distribution of features extracted from sensor readings is different across individuals, which may drop the performance of the model trained with multiple individuals.

In this experiment, we extract all time-domain features for the machine learning model following the method [27], including mean, std, etc. We then simply use a Random Forest (RF) with default settings as the classification algorithm. Three types of models are designed to compare the performance: the **generic model**, **Transfer model**, and **personal model**. The different models are described as follows:

**Generic model**: The data from all individuals are packed into one large dataset, which is divided into training and test sets. Namely, the model is trained and tested on the dataset with the data from all of the individuals.

**Transfer model**: We follow the settings of transfer learning, where the model is trained on a certain single individual and then tested on the dataset from another individual. Please note that performance is average by all source individuals and each individual is randomly selected.

**Personal model**: The data from each individual are divided into training and test sets. The model is trained using the data from only one single individual and then tested on the same individual. Note that the performance is averaged across all individuals.

We use the five-fold cross-validation method in parameter searching and while training the models. We classify the activity annotations into three types, which are activities, locations, and social relations. The performances of the three types of activity are displayed in Figure 2, where each set of the bars represents the $f_{1}score$ of recognizing the activity (blue bar), location (red bar), and social relation (green bar) for each individual. Results indicate that the personal model performs the best and the transfer model has the worst performance, which indicates that heterogeneity exists among individuals. This experiment shows that model personalization improves the performance in the activity recognition task in an open-world environment. Thus, using federated learning to learn features from multiple individuals and generating a model for each individual by meta-learning is a promising approach to account for inter-individual differences.

To conclude, the result of an empirical experiment on the activity recognition model indicates that distribution discrepancy across individuals should be considered. Thus, we apply federated learning to learn general features from multiple client users and generate adaptive models for each type of individual. In addition, the fact that different individuals have different feature distributions motivates the use of attention mechanisms. However, it is impractical to train one single model for each individual in the real life. In addition, the amount of a single individual’s dataset is limited for a machine learning model, especially for deep neural networks. Therefore, we intuitively account for the heterogeneity of the features and, meanwhile, utilize the shared knowledge transferred across individuals.

## 4. Methodology

This section describes the method of the federated meta-learning framework for diversity-aware activity recognition. Figure 3 illustrated the overall architecture of our approach. First of all, individuals are clustered according to their characteristics including basic information, personality, and mental state. Then, an activity recognition model is trained locally in a federated learning manner. Finally, the diversity-aware activity recognition model is generated from the central server. In what follows, the detailed methodology will be introduced.

### 4.1. Problem Definition

In order to define our federated meta-learning framework for diversity-aware activity recognition, we give its definition as follows: The set of multi-modal sensor readings is denoted as: $x=\left\{S_{k}\right\},k\in \left\{1,\ldots ,K\right\}$. Each sequence of single-modality sensor data from the smartphone is defined as: $S_{k}^{t}=\left\{s_{k}^{1},s_{K}^{2},s_{t}^{3},\ldots ,S_{k}^{t},\right\},t\in T$, where *T* refers to the length of the time window of sensory readings. U indicates the set of individuals. For each such individual u∈U, we have access to a corresponding data set $D_{u}={\left\{\left(x_{i},y_{i}\right)\right\}}_{i=1}^{n_{u}}$, where $x_{i}\in R^{d}$ are sensor readings and $y_{i}$ are the corresponding ground-truth activity labels.

### 4.2. Overall Architecture

In the federated learning framework, we aim to train a meta-model with decentralized data from and under the coordination of a centralized parameter server. In order to achieve strong generalization across diverse individuals, we consider both heterogeneity and similarity between individuals by leveraging a Federated meta-learning architecture, as shown in Figure 4. The proposed architecture consists of a central model, with parameter $\Theta_{c}$, and *m* decentralized models $W_{u},u\in \left\{1,2,\cdots ,m\right\}$ that learn cluster-specific features. The overall goal is to acquire a diversity-aware activity recognition model that generalizes across multiple clusters of individuals, represented by U. For now, let us focus on the first desideratum. This can be implemented by minimizing an appropriate loss over the observed clusters of individuals, as follows:(1)$$\min_{x\in R^{d}}f\left(x\right):=\frac{1}{m}\;\sum_{u=1}^{m}F_{u}\left(x_{u};\Theta_{c},W_{u}\right)$$
where $F_{u}\left(x_{u};\Theta_{c},W_{u}\right)≜E_{\xi \;D_{u}}\left[F_{i}\left(x_{u},\xi_{u}\right)\right]$ denotes the loss function of local client model, which essentially evaluates the average discrepancy between the output of client model $f_{u}\left(\cdot \right)$ and the ground truth corresponding to a random training sample $x_{u}$ that follows a local data distribution $\xi_{u}$. The parameter *d* represents the dimensionality of the training model and $f_{u}\left(\cdot \right)$ denotes the activity recognition model in each client that depends on both the shared parameters $\Theta_{c}$ and the cluster-specific parameters $W_{u}$. The architecture of the model is detailed next.

### 4.3. Individual Clustering

We study the factors that impact individuals’ behavior patterns or lifestyles based on previous work on mobile sensing and sociology analysis. Existing work has proven that personality [22,24,66] and psychological factors [25,26] have correlations with behavioral patterns. Based on this observation, we assume that the same activity can be performed differently by individuals due to their diverse social characteristics such as personalities and emotional states. Table 1 shows the features for clustering individuals including social factors and behavioral factors. In addition, we analyze the features of the behavioral patterns. Therefore, we cluster individuals based on these factors. In the data collection procedure, the Big Five trait taxonomy [67] and psychological factors are collected. We then apply K-means [68] to cluster individuals according to their personality traits and mental health questionnaires.

The features in the first group capture information about questionnaires about individuals as follows:**Basic Information:** We leverage basic information of individuals (e.g., age and department of university) to cluster individuals. Note that the data are anonymized, and informed consent was signed to inform individuals of privacy and ethics.**Personality:** We account for personality to cluster individuals because the research carried out by researchers [22] indicates that personality traits can reflect the key parts of how a person thinks, feels, and thus behaves. We applied a Big Five (i.e., extroversion, agreeableness, openness to experience, conscientiousness, neuroticism) questionnaire, which is a widely accepted personality measurement tool.**Mental State:** We also applied the mental state information of individuals for clustering. Specifically, we applied the PHQ-9 questionnaire, which is a subset of 9 questions based on DSM-IV (Diagnostic and Statistical Manual of Mental Disorders) criteria [69]. It is a simple and effective self-rating scale for depression with high reliability and validity.

The second group of features is concerned with behavioral factors of individuals as follows:**Behavior type:** We account for the types of activity labels to cluster individuals. To be specific, we cluster individuals who tend to have similar types of activity labels together.**Diversity of behavior pattern:** By diversity, we mean the intrinsic heterogeneity in its activity patterns inspired by the study on human behavior [15]. For each individual *u*, we measure this by computing the number of distinct activities that they perform, and the Shannon entropy of the activity annotations $\left\{y_{i,a}\right\}$ available in the training set. These meta-features model the intrinsic difficulty of predicting the behavior of an individual *u* and are useful for preventing unpredictable individuals to be used as sources.

### 4.4. Feature Representation Network

The feature representation network learns the features from sensor observations *x* using a CNN-RNN architecture, together with a cluster-specific mask based on the attention mechanism, which is shown in Figure 3. As inputs for the neural networks, the training instances are partitioned by the fixed-size sliding window into *k* time intervals of length *L*. This results in a data matrix of shape $d_{s}\times L$, where $d_{s}$ is the dimension for each sensor *s* (e.g., x, y, and z axes for the accelerometer). We then apply a Fourier transform to compute frequency-domain information, obtaining a final input tensor $X_{s}$ of shape $d_{s}\times 2f\times k$, where *f* is the dimension of frequency-domain information. The set of tensors for each sensor, $X=\{X_{s}\}$, is the final input of the embedding network.

The embedding network itself uses two sets of convolutional layers: the first set is applied to each sensor separately, and the second one is applied to the concatenation of the individual sensor embeddings (see Figure 3), so as to fuse their representations and extract spatial dependencies between them. Within the two CNNs, we apply an attention-based mask to extract individual-specific features, which will be introduced in Section 4.5. Then, Gated Recurrent Unit (GRU) layers are used to extract the temporal relevance of the *k* CNN outputs. Finally, the embedding vectors output by the GRU layers is fed to a fully connected output layer that computes the probabilities for each category using a softmax activation. Note that we generate a fully connected layer adaptively according to the types of activities performed by the cluster of individuals, considering the fact that different groups of individuals perform diverse kinds of activity.

### 4.5. Diversity-Aware Attention-Based Mask

As mentioned above, the recognition of different users’ activities relies on the different sensor readings. In order to precisely adapt the central model to a specific cluster of users, we apply the attention-based mask to the feature representation layers, aiming at extracting user-specific information. Therefore, we train multiple user-specific attention networks. As such, the attention masks can be considered as feature selectors from the shared network, while the shared networks can learn shared features across all users. Recall that our embedding network contains two types of CNN layers: multiple sensor-specific convolutional layers and a fusion convolutional layer. We apply the attention module to both of the two types of convolutional layers, as shown in Figure 5.

The detailed structure of the attention-based mask is shown in Figure 5, consisting of multiple convolutional blocks for extracting task-specific features. Specifically, we refer to the shared features in the *l*-th layer of the shared network as $e^{l}$ and the learned attention mask in this layer for individual *u* as $e_{u}^{l}$. The task-specific features ${\hat{e}}_{u}^{l}$ in this layer are then computed by element-wise multiplication of the attention masks with the shared features:(2)$${\hat{e}}_{u}^{l}=Mask_{u}^{l}⊙p^{j}$$

For the first attention module in the convolutional layers, we take as input only features in the shared network. As for the subsequent attention mask in layer *j*, the input is the concatenation of the shared features $p^{j}$ and the task-specific features from the previous layer ${\hat{a}}_{i}^{j-1}$:(3)$$Mask_{u}^{l}=h\left(g\left(\left[p^{l};f\left({\hat{e}}_{u}^{\left(l-1\right)}\right)\right]\right)\right),$$
Here, f,g,h are convolutional layers with batch normalization, following a non-linear activation ReLu in f,g or Sigmoid in *h*. Both *f* and *g* are composed with a [3×3] kernel, while *h* has a [1×1] kernel to match the channels between the concatenated features and the shared features. Then, the attention masks $Mask_{u}^{l}\in \left[0,1\right]$ is learned with back-propagation, which can operate as feature selectors from the shared features, while the shared network learns generalized features across all individuals.

### 4.6. Implementation for Federated Model Training

Recall that our model is split into shared and cluster-specific parts, which are stored separately. The central model $\Theta_{c}$ contains two CNNs that perform single-sensor feature extraction and multiple sensors feature fusion. As for the decentralized individual models, $W_{u}=\left\{a_{u},h_{u},c_{u}\right\}$, where $a_{u}$ indicates attention-based mask modules for extracting cluster-specific features, $h_{u}$ indicates a GRU module for extracting temporal features, and $c_{u}$ refers to output layer for classification. In this way, both user-agnostic and cluster-specific features can be extracted by the proposed framework. The training procedure is shown in Algorithm 1. To optimize the parameter and update the model, the parameters are transferred between the central server and distributed clients. Specifically, each user with a local dataset $D_{u}$ obtains CNN models $\Theta_{c}$ from the central server and introduces their data into the CNNs masked by their local attention module to obtain their specific feature embeddings. Then, embedding vectors are introduced to GRU to obtain temporal features and finally obtain the loss via the classification layer. By performing *n* epochs of training locally in the clients, the parameters are separately updated to a central server and decentralized nodes. The central model then averages the updated parameters to update the shared embedding network in the central model by averaging the models.
**Algorithm 1:** Federated Multi-Task Attention for Diversity Mental activity recognition.1:**Input**: *m* individual-specific data sets $\left\{D_{u}\right\}$, one per client.2:**Output**: central model $\Theta_{c}$, individual-specific models $\left\{W_{u}\right\}$.3:**# Training Central Model:**4:Initialize central model $\Theta_{c}\leftarrow \Theta_{0}$5:**for** round=1,2,⋯**do**6:     **for** each u∈{1,2,⋯,m} in parallel **do**7:        Get central model $\Theta_{c}$ from the server.8:        Train for *n* epochs using central model $\Theta_{c}$ together with local model $W_{u}$, and get locally updated parameters $\Theta_{u}$ and $W_{u}$.9:        Push updated parameters $\Theta_{u}$ to server.10:   **end for**11:   Update $\Theta_{c}$ according to $\Theta_{c}=\Theta_{c}+\lambda \left(\hat{\Theta }-\Theta_{c}\right)$, where $\hat{\Theta }=\frac{1}{m}\sum_{u=1}^{m}\Theta_{u}$12:**end for**13:**return** $\Theta_{c}$ and $\left\{W_{1},\ldots ,W_{m}\right\}$14:**# Model Personalization:**15:**for** user *u* in all users **do**16:    Pull parameters of embedding network $\Theta_{c}$ from FL server.17:    Fine-tune $\Theta_{c}$ with pairwise loss on the local dataset to obtain local embedding network $\Theta_{u}$.18:    Further fine-tune local classifier $W_{u}$ with cross-entropy loss on local dataset.19:    **return** personalized classification model $W_{u}$ for user *u*.20:**end for**

## 5. Data Collection

To evaluate the effectiveness of the model in the real world, We construct two activity recognition datasets involving multiple individuals, which aim at studying the behavior pattern and lifestyle of university students, supported by *anonymized* projects. The collection and construction procedure is similar to the *anonimized* dataset and uses exactly the same tools and techniques collected as *anonymized*. During the data collection procedure, a smartphone app was used to carry out sensor recording (e.g., GPS, accelerometer) and administer periodic questionnaires about activity, location, and social activity. All students signed informed consent forms. The main features of this dataset are that it: (1) contains annotations for complex activities such as “Housework” and (2) is collected “in the wild” in an unconstrained setup. In this experiment, the records are annotated with multiple types of activity, including activities, locations, and social relations. The signals are obtained from smartphone sensors and include motion-reactive sensors (e.g., accelerometer), location, phone state, etc. Specifically, the students enrolled in the university who were interested in the data collection pilot were invited to an introductory presentation where they received the basic information about the project and the aims of the pilot. Note that informed consent was signed to inform the students of privacy and ethics. Considering the duration of the data pilot is two weeks, the participants were allowed to quit at any time during the pilot.

### 5.1. Annotated Sensor Data Collection

The pilot relied on a smartphone app which provided sensor data collection and time diaries. All the participants were required to install the app on their smartphones, which recorded streaming data from both hardware (e.g., GPS, accelerometer) and software (e.g., running applications). The full list of sensors and their sampling rate is shown in Table 2.

In addition to the sensor data, a self-report time diary composed of three questions on activities, locations, and social relations was asked by the application every 30 min. The questions are designed according to [70], which aims at studying where the time of participants is spent each day. Table 3 shows the questions and their answers generated during the pilot. Note that only one answer can be selected.

### 5.2. Personality and Mental State Survey

To learn the traits of participants, at the beginning of the pilot, the participants were required to finish a series of entry questionnaires including the Big Five personality traits [71], health awareness, and the Patient Health Questionnaire (PHQ-9) [69]. The Big Five (i.e., extroversion, agreeableness, openness to experience, conscientiousness, neuroticism) questionnaire is a widely accepted personality measurement tool that reflects the key parts of how a person thinks, feels, and behaves. PHQ-9 is a subset of 9 questions based on DSM-IV (Diagnostic and Statistical Manual of Mental Disorders) criteria [69]. It is a simple and effective self-rating scale for depression with high reliability and validity.

### 5.3. Data Exploration

To construct the dataset for machine learning, we extract three annotations about the activity as labels of the multiple sensor readings as in the settings of the previous research, *anonymized*. Specifically, the input data are composed of multiple 30 min sequential data which include the 15-min periods before and after answering the questions. In order to utilize more precise information, we use only the central 10 min sequence (5 min before and 5 min after answering the questions) as the input of the model. We then perform data segmentation to split the data into 60 10 s windows, following similar settings and lengths as existing benchmark datasets [72,73]. As for the label, activity labels annotated are extracted as the ground-truth labels of the sensor sequences. Note that we use only activity-related labels, considering that these activity labels have stronger correlations with sensory data and thus are more suitable for the task of activity recognition, following most studies on Human Activity Recognition (HAR) [34,72,74]. After data cleaning and preprocessing, *dataset1* contains 30 individuals and *dataset2* contains 48 individuals. The frequency of behavioral activity annotations is shown in Figure 6.

## 6. Evaluation

In this section, numeric experiments are set up to evaluate our proposed model to answer the following questions:**Q1:** Does DivAR handle inter-individual heterogeneity effectively for the activity recognition system?**Q2:** Does DivAR have the ability to adapt the model to new-coming users to address the “cold-start” problem for the activity recognition system?**Q3:** What features can be used to cluster individuals properly to achieve high-performance, diversity-aware activity recognition?**Q4:** How does DivAR perform with different hyperparameters and when the computation is complex?

### 6.1. Experiment Settings

We use both two datasets to evaluate the effectiveness of our proposed model. In addition, we create a third dataset by mixing the two datasets, which are more heterogeneous and contain more individuals. We apply the settings of meta-learning by splitting all users into meta-train users, which participate in the meta-learning process, and meta-test users, which serve as new users for testing the generalization ability of the meta-learned model. To be specific, we randomly select one user as the meta-test user and the rest as meta-train users to evaluate the performance for the proposed method and all baselines. We repeat the whole process 10 times and average the performance. To train the model and test the performance, we further split the dataset of each individual into a training set (80%) and a test set (20%). Considering that the labels are imbalanced, we use both macro−F1 and accuracy as the performance metrics in the evaluation. The number of clusters is selected as five.

We implemented our model using Python 3.6 and Pytorch 1.8. All experiments are carried out on a machine with two NVIDIA GeForce RTX 3090 GPUs. The Adam optimizer [75] with $\beta_{1}=0.9$, $\beta_{2}=0.98$, and $\varepsilon =10^{-8}$ is used to update all network parameters. In the federated learning procedure, we set λ=1.0 and perform n=5 epochs of local training in each update round. The time window of input for the CNN is selected as 2 s. Considering that the labeled data are typically imbalanced, we apply both accuracy and macro-average $f_{1}score$ for evaluation.

### 6.2. DivAR Can Handle Inter-Individual Heterogeneity Effectively for Existing Users in the Activity Recognition System

To statistically measure differences in performance, we select the following methods for comparison. Specifically, the task of activity recognition is essentially the problem of sensory time series classification; thus, we compare our model with sensory time series classification models. Considering the fact that most existing work on activity recognition targets recognizing activity, we compare state-of-the-art models on HAR. In addition, we take the meta-learning and federated learning model into consideration for fair and extensive evaluation. The description of baseline models is listed as follows:**Random Forest** [76]: The Random Forests are a type of ensemble classification model constructing multiple decision trees. Note that we extract features from sensory data manually for training the Random Forest classifier.**DeepSense** [34]: A state-of-the-art model using CNN-RNN structure for several types of multimodal sensor series classification.**AttenSense** [74]: An attention-based multimodal neural network model for multimodal sensor series classification.**Meta-HAR** [21]: A federated representation learning framework, in which a signal embedding network is meta-learned in a federated manner and personalized models are adapted for each user.**DivAR-individual**: Our proposed federated meta-learning model without individual clustering, training personalized models for individuals.**DivAR-cluster**: Our proposed federated meta-learning model with individual clustering according to different features, training a cluster-specific model for each cluster of individuals.

We first evaluate the effectiveness of generalizing within existing users in an activity recognition system. Table 4 shows the performance attained by our proposed DivAR and other activity and activity recognition algorithms on both of the target datasets and on the mixing dataset over ten independent runs. As reported in Table 4, our proposed DivAR consistently outperformed the baseline activity and activity recognition model both in mocro−F1 score and accuracy on the three datasets, averaged by ten independent runs. The improvements are statistically consistent and significant on the three datasets with performance gain up to 6.87% in accuracy and 8.44% in mocro−F1 between DivAR and baseline models.

DivAR is better than AttenSense, which shows that the attention weights on multimodal sensors cannot be shared among all individuals and that each individual should be allocated different attention because of the diverse behavior patterns. DivAR performs better than Meta-HAR, which shows that heterogeneity should not only be considered in the classification layer as it also exists in the feature extraction procedure. Comparing the performance between the two versions of our proposed model, DivAR-cluster outperforms DivAR-individual consistently on the three datasets and significantly on the mixing dataset, with performance gains up to 5.50% in accuracy and 9.61% in mocro−F1. This indicates that clustering individuals can enhance the generalization ability by considering individuals’ characteristics. For detailed performance on each label, the confusion matrix on each cluster is shown in Figure 7.

### 6.3. DivAR Can Be Adapted to New Users Effectively

In order to address the ‘cold-start’ problem for the activity recognition system, we simulate the situation when a new user comes to the system with a few labeled data. Namely, we generate a new model for the new user, with a central embedding network and multiple existing cluster-specific models. The results are analyzed and discussed as follows.

Table 5 shows the performance attained by our proposed DivAR and other activity and activity recognition algorithms on all three target datasets. We randomly select ten individuals and use the mixing dataset over ten independent runs. As indicated in the Table, DivAR generally outperforms other methods on four datasets, which means that it can handle datasets with high heterogeneity effectively and can easily adapt to new individuals. Generally, the deep learning model can extract features and generalize to new users better than conventional shallow models. The fact that DivAR performs better than Meta-HAR shows that new individuals tend to have discrepancies with existing users in terms of feature representation and that user-specific attention modules can learn diverse features better. Comparing the performance between DivAR-cluster and DivAR-individual, in most of cases, DivAR-cluster outperforms DivAR-individual on the three datasets. However, the performance gain of the clustering version is not as significant as the one featuring existing individuals (in Table 4). This indicates that clustering individuals can enhance the generalization ability by considering individuals’ characteristics, but it is not always necessary.

### 6.4. Evaluation on Features for Clustering

To understand the contribution of the proposed attention modules, we visualize the attention masks of the sensor fusion layer for each sensor across multiple clusters of individuals. As shown in Figure 8, the different weights of various sensors are learned by our proposed approach. In particular, the attention masks have strong diversity across clusters of individuals, which validates the argument of the motivating example in Section 3.

We then evaluate the performance of the models using different features to cluster individuals. As shown in Table 6, we evaluate the two types of clustering methods: using extra information and using labeled data. Note that model generalization indicates the task of generalizing the model to existing individuals, while model adaptation refers to adapting the model to new users. Generally speaking, the performances of using each kind of feature are relatively stable. Surprisingly, clustering individuals using their behavior type information performs the best. We assume that using label information, such as behavior type, works because individuals who have the same types of activities tend to have similar marginal and conditional distributions of their sensory features. However, it is difficult and expensive to acquire label information from new users; thus, more distribution features from sensory data should be explored for the model’s deployment in real-world scenarios.

### 6.5. Computational Complex Analysis

In this subsection, we analyze the computational complex analysis by comparing DivAR-cluster and DivAR-individual. Specifically, we check how accuracy changes when we increase the number of client training epochs from 5 to 25. The results, illustrated in Figure 9, verify that DivAR-cluster consistently outperforms DivAR-individual on both datasets. The performances of both DivAR-cluster and DivAR-individual are relatively stable with the change in training epochs. Moreover, DivAR generally takes fewer epochs to achieve its best performance. This further stresses the effectiveness of our approach. In addition, we compute the number of parameters. DivAR model contains 300 k parameters for each cluster, which does not cost much time to run on mobile devices.

### 6.6. Discussion

Compared to recent popular deep learning methods for human activity recognition, DivAR is a meta-learning approach with a federated learning architecture, and it can improve the performance on HAR compared to state-of-the-art traditional methods. The DivAR model can extract features and be generalized to new users better than the state-of-the-art models. In addition, clustering individuals can enhance the generalization ability by considering individuals’ characteristics. As for computation cost, DivAR generally takes fewer epochs to achieve its best performance.

One possible limitation of DivAR may be that it relies on deploying deep learning models on devices such as smartphones and wearables, which requires large computational resources from edge devices. Thus, deep learning methods often must be compressed manually for small devices. In addition, more unsupervised methods for clustering users should be applied to avoid collecting information from users.

## 7. Conclusions

To address privacy issues and distribution divergence in activity recognition in real-world scenarios, we propose DivAR, a diversity-aware activity recognition framework based on a federated Meta-Learning model which can extract individual-agnostic sensory features using a centralized embedding network and individual-specific features using a attention module embedded in each decentralized model. Our proposed framework first classifies individuals into multiple clusters according to their behavioral patterns and social factors. We then apply meta-learning in the architecture of federated learning, where a centralized meta-model learns common feature representation that can be transferred across all clusters of individuals and multiple decentralized cluster-specific models are utilized to learn cluster-specific features. For each cluster-specific model, a CNN-based attention module learns cluster-specific features from the global model. In this way, by training the model with sensory data locally, privacy-sensitive information existing in the sensory data can be preserved. Finally, to evaluate the model, we conducted experiments on the two datasets, which are collected from multiple individuals in real-world scenarios. The results show that the proposed diversity-aware activity recognition model has a relatively better generalization ability than other models and achieves competitive performance in multi-individual activity recognition tasks. As for future work, more social factors will be explored to improve the performance of our model; meanwhile, correlation analysis will be carried out to enhance the explainability of the model.

## Figures and Tables

**Figure 1 sensors-23-01083-f001:**
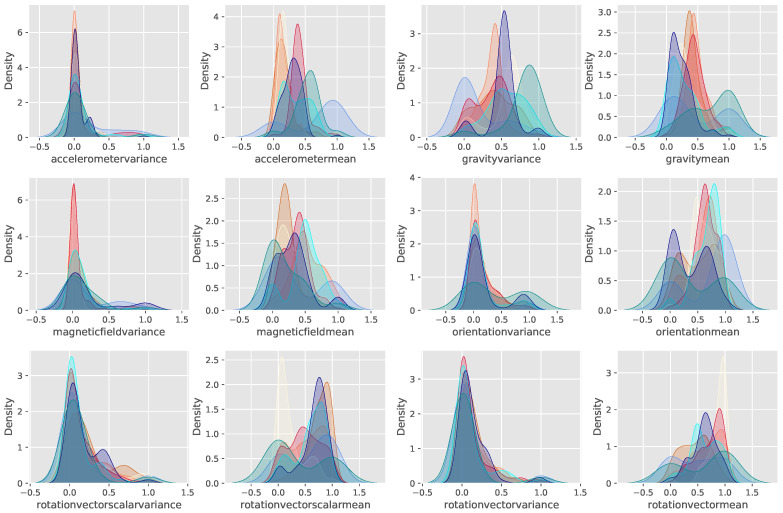
Distribution of each feature across individuals. Each figure refers to the distribution of a certain feature, and the curves in each subplot are the distribution of the feature estimated by the KDE of eight randomly selected individuals.

**Figure 2 sensors-23-01083-f002:**
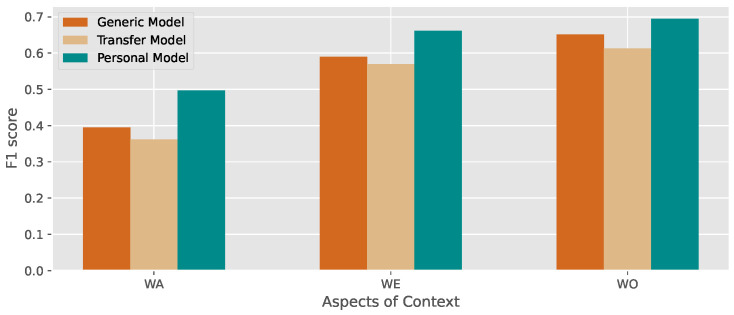
Comparison of the performance of the generic model, personal model, and transfer model.

**Figure 3 sensors-23-01083-f003:**
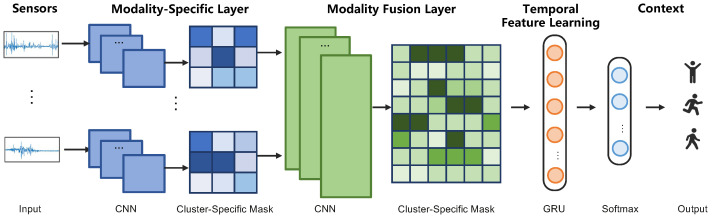
The architecture of the attention-based meta-learning model for diversity-aware activity recognition. The CNNs in the modality-specific layer and modality fusion layer are transferred from the central model and trained locally with the rest components on the client side. The detailed structure of the attention-based, cluster-specific mask operation is introduced in the next subsection.

**Figure 4 sensors-23-01083-f004:**
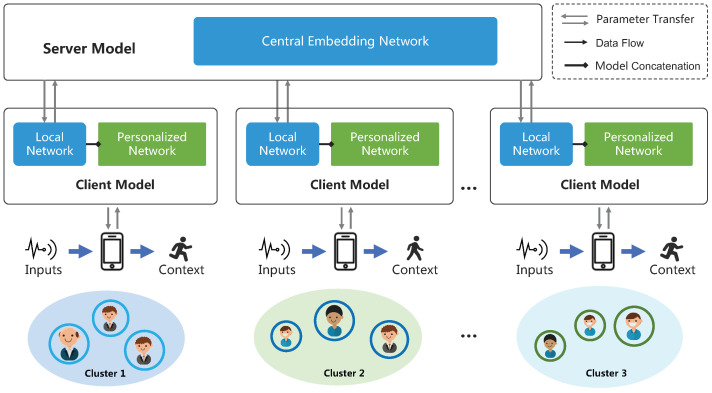
Overall architecture of the diversity-aware activity recognition.

**Figure 5 sensors-23-01083-f005:**
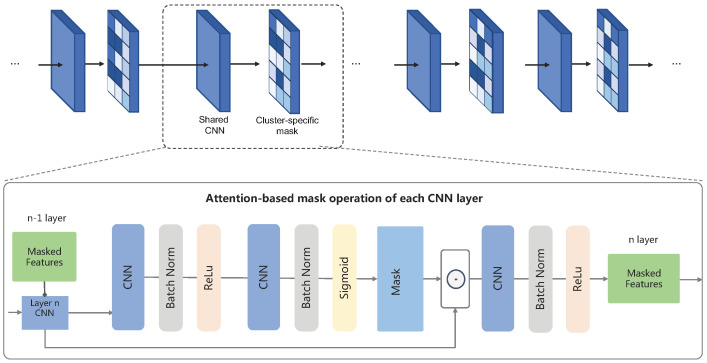
Architecture of the attention-based mask for cluster-specific feature extraction.

**Figure 6 sensors-23-01083-f006:**
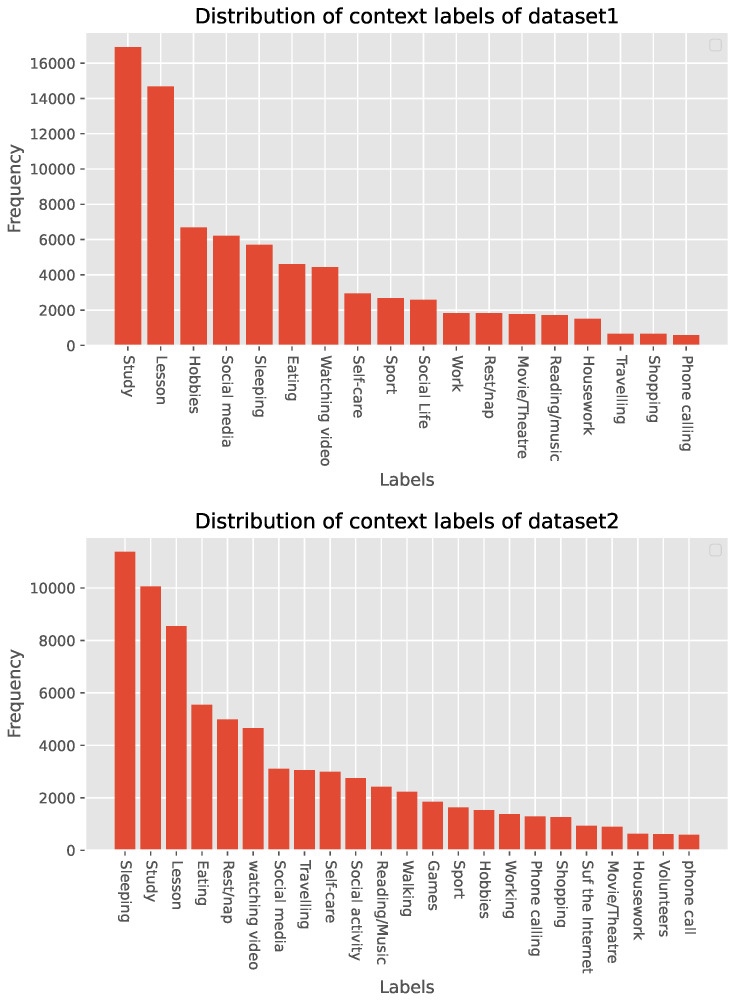
Frequency of activity annotations.

**Figure 7 sensors-23-01083-f007:**
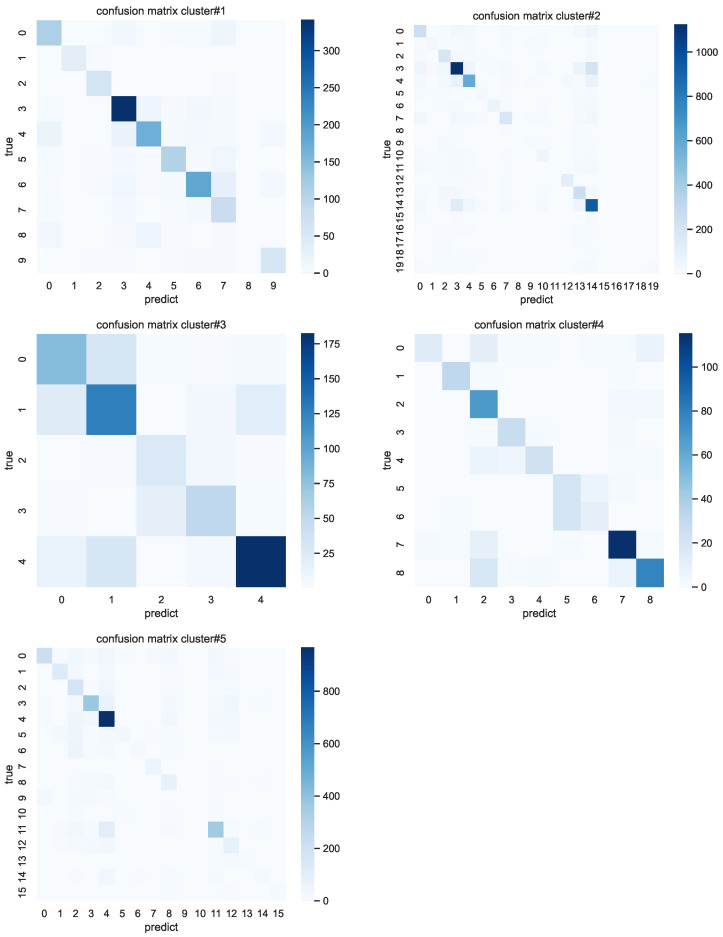
Confusion matrix of each cluster.

**Figure 8 sensors-23-01083-f008:**
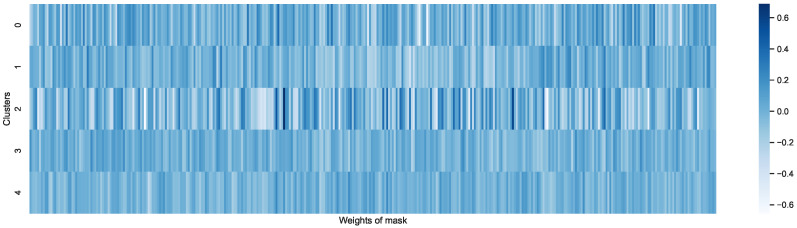
Weight visualization of diverse clusters.

**Figure 9 sensors-23-01083-f009:**
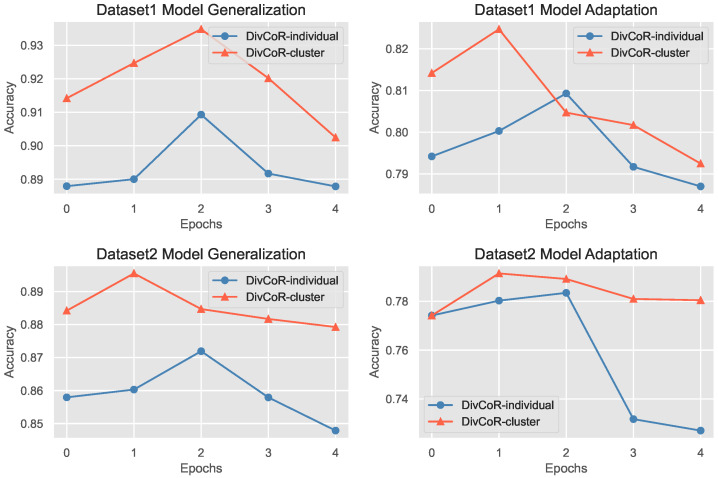
Evaluation on local train epochs.

**Table 1 sensors-23-01083-t001:** List of features for clustering individuals. In the Measurement column, $p_{u,a}$ indicates the probability of activity *a*.

Feature Type	Feature	Measurement
Social factors	Basic Information	Year of birth, dormitory, grade, department
	Personality	Big Five questionnaires
	Mental state	PHQ-9 questionnaires
Behavioral factors	Behavior type	Types and numbers of activity
	Diversity	Shannon entropy of annotations: $-\sum_{a}p_{u,a}ln\left(p_{u,a}\right)$

**Table 2 sensors-23-01083-t002:** List of sensors.

Sensor	Sampling Rate	Unit
Acceleration	20 Hz	m/s${}^{2}$
Linear Acceleration	20 Hz	m/s${}^{2}$
Gyroscope	20 Hz	rad/s
Gravity	20 Hz	m/s${}^{2}$
Rotation Vector	20 Hz	Unitless
Magnetic Field	20 Hz	μT
Orientation	20 Hz	Degrees
Temperature	20 Hz	°C
Atmospheric Pressure	20 Hz	hPa
Humidity	20 Hz	%
Proximity	On change	0/1
Position	Every minute	Lat./Lon.
WIFI Network Connected	On change	Unitless
WIFI Networks Available	Every minute	Unitless
Running Application	Every 5 s	Unitless
Screen Status	On change	0/1
Flight Mode	On change	0/1
Battery Charge	On change	0/1
Battery Level	On change	%
Doze Modality	On change	0/1
Headset plugged in	On change	0/1
Audio mode	On change	Unitless
Music Playback	On change	0/1
Audio from the internal mic	10 s per minute	Unitless
Notifications received	On change	Unitless
Touch event	On change	0/1
Cellular network info	Once every minute	Unitless

**Table 3 sensors-23-01083-t003:** The questionnaire for collecting activity labels.

Q1. What Are You Doing?	Q2. Where Are You?	Q3. With Whom Are You?
Sleeping	Home Apartment Room	Alone
Self-care	Relatives Home	Friend(s)
Eating	House (friends others)	Relative(s)
Study	Classroom / Laboratory	Classmate(s)
Lesson	Classroom / Study hall	Roommate(s)
Social Life	University Library	Colleague(s)
Watching iQiyi, Youku, Bilibili, etc.	Other university places	Partner
Social media (QQ, WeChat, Weibo, etc.)	Canteen	Other
Traveling (*)	Other Library	
Coffee break, cigarette, beer, etc.	Gym	(*) How are you moving?
Phone calling; in chat QQ or WeChat	Shop, supermarket	By subway
Reading a book, listening to music	Pizzeria, pub, bar, restaurant	By car
Movie Theatre Concert Exhibit	Movie Theater, Museum	By foot
Housework	Work place	By bike
Shopping	Other place	By bus
Sport	Outdoors	By train
Rest/nap		By motorbike
Hobbies		Other
Work		

**Table 4 sensors-23-01083-t004:** Performance (Accuracy and $F_{1}score$) on generalizing within existing individuals of the baseline method and DivAR on three datasets.

Model	Dataset1		Dataset2		DatasetMix	
	Accuracy	F1-Score	Accuracy	F1-Score	Accuracy	F1-Score
Random Forest	0.6814	0.6523	0.6428	0.5971	0.5385	0.5246
DeepSense	0.8141	0.7879	0.8136	0.7405	0.7123	0.6430
AttenSense	0.8369	0.8194	0.8014	0.7580	0.7321	0.7088
Meta-HAR	0.8661	0.8193	0.8424	0.8186	0.8021	0.6989
DivAR-individual (ours)	0.9021	0.8780	0.8711	0.8017	0.7848	0.6995
DivAR-cluster (ours)	**0.9348**	**0.9037**	**0.8955**	**0.8319**	**0.8395**	**0.7836**

**Table 5 sensors-23-01083-t005:** Performance (Accuracy and $F_{1}score$) of baseline method and DivAR on adapting to a new user on three datasets.

Model	Dataset1		Dataset2		DatasetMix	
	Accuracy	F1-Score	Accuracy	F1-Score	Accuracy	F1-Score
Random Forest	0.6128	0.5785	0.6421	0.5972	0.5122	0.4319
DeepSense	0.6523	0.5107	0.6881	0.800	0.6231	0.813
AttenSense	0.7062	0.6492	0.7012	0.6082	0.5832	0.5481
Meta-HAR	0.7319	0.709	0.7379	0.6618	0.6227	0.5740
DivAR-individual (ours)	0.8093	0.7205	0.7835	**0.7602**	0.7098	0.6811
DivAR-cluster (ours)	**0.8247**	**0.7479**	**0.7901**	0.7519	**0.7308**	**0.7092**

**Table 6 sensors-23-01083-t006:** Performance (Accuracy and $F_{1}score$) on two tasks using different features to cluster on Dataset1. MG indicates model generalization and MA represents model adaptation.

Tasks	All Features	Basic Info	Personality	Mental State	Behavior Type	Diversity
MG (Acc)	0.9348	0.9216	0.9198	0.9273	**0.9421**	0.9012
MG (F1)	0.9037	0.8723	0.9012	0.8927	**0.9176**	0.8628
MA (Acc)	**0.8247**	0.8128	0.8091	0.8201	0.8201	0.8172
MA (F1)	0.7479	0.7311	0.7190	0.7201	**0.7589**	0.7077

## Data Availability

Not applicable.

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
