# Peer review of "Federated Meta-Learning with Attention for Diversity-Aware Human Activity Recognition"

_sensors, 2023, doi:10.3390/s23031083_

Round 1

Reviewer 1 Report

The work is interesting. This paper has merit and covers an important topic; however, I have the following suggestions to improve the quality of this paper:

 -There are so many grammatical/typos errors. For instance: Line no. 146, We firstl exploreèWe first explore,

There are many more such errors; revise the manuscript.

 - Somewhere figures are written as Fig, and some of them are Figure. Follow the unique style.

- computational complex analysis may be added

 - Add a discussion section before conclusion

I would strongly encourage the authors to share a prototype implementation of their algorithm for the benefit of the greater community.

Author Response

Please find the replies attached.

Reviewer 2 Report

The authors describe a proposed activity recognition framework utilizing federated learning, that handles the issue of divergence in activity recognition of multiple subjects. They also constructed two heterogeneous datasets for AR.  The authors first cluster the subjects, then utilize their framework and they evaluate it on two data collections they created. Abstract and introduction are very well written and outline the work and contribution of the paper clearly.

The proposed methodology is interesting and is compared with other ML algorithms and NNs. 

There are two points the authors may consider:

-if the datasets they collected will be publicly available

-why the related work section is in the end of the paper

Author Response

Please find the replies attached.
